# Polyphenolic Compounds in Fabaceous Plants with Antidiabetic Potential

**DOI:** 10.3390/ph18010069

**Published:** 2025-01-09

**Authors:** Lucia Guerrero-Becerra, Sumiko Morimoto, Estefania Arrellano-Ordoñez, Angélica Morales-Miranda, Ramón G. Guevara-Gonzalez, Ana Angélica Feregrino-Pérez, Consuelo Lomas-Soria

**Affiliations:** 1Center of Applied Research in Biosystems (CARB-CIAB), School of Engineering, Autonomous University of Querétaro-Campus Amazcala, Carr. Amazacala-Chichimequillas km 1.0, El Marqués, Querétaro 76265, Mexico; gubel28@gmail.com (L.G.-B.); eaordonez21@gmail.com (E.A.-O.); ramon.guevara@uaq.mx (R.G.G.-G.); 2Research and Postgraduate Division, School of Engineering, Universidad Autónoma de Querétaro, Campus Amazcala, Carretera a Chichimequillas Km 1 s/n, Amazcala, El Marqués, Querétaro 76265, Mexico; 3Departamento de Biología de la Reproducción, Instituto Nacional de Ciencias Médicas y Nutrición Salvador Zubirán, Ciudad de México 14080, Mexico; sumiko.morimotom@incmnsz.mx (S.M.); angelica.moralesm@incmnsz.mx (A.M.-M.)

**Keywords:** antioxidant, diabetes, legumes, secondary metabolites, phytochemicals

## Abstract

Diabetes mellitus (DM) is a chronic non-communicable disease with an increasing prevalence in Latin America and worldwide, impacting various social and economic areas. It causes numerous complications for those affected. Current treatments for diabetes include oral hypoglycemic drugs, which can lead to adverse effects and health complications. Other natural alternatives for DM treatment have been studied as adjunct therapies that could reduce or eliminate the need for antidiabetic medications. Several natural supplements may offer an alternative way to improve the quality of life for patients with DM, and they may have other nutraceutical applications. Due to their phenolic compound content, some leguminous substances have been proposed as these alternatives. Phenolic compounds, with their high antioxidant activity, have shown promising potential in insulin synthesis, secretion, and the functionality of the endocrine pancreas. This review provides valuable information on various leguminous plants with anti-diabetic properties, including antioxidant, hypoglycemic, anti-fat-induced damage, and anti-apoptotic properties in vitro and in vivo, attributed to the high content of phenolic compounds in their seeds. Natural products with antidiabetic and pharmacological treatment potential improve diabetes management by offering more effective and complementary alternatives. To integrate these herbal remedies into modern medicine, further research on phenolic compound type, doses, efficacy, and safety in the human population is needed.

## 1. Introduction

Diabetes mellitus (DM) is a non-communicable disease characterized by high blood glucose levels (hyperglycemia), with disturbances in carbohydrate, fat, and protein metabolism. These metabolic disorders are caused by defects in insulin secretion and/or a lack of response to the action of this hormone in target organs. When DM is not controlled correctly, it can result in severe complications, including cardiovascular disease, renal failure, nerve damage, and blindness, as well as early death [1,2,3]. In treating DM, several therapeutic measures are fundamental for adequate metabolic control. These include some non-pharmacological measures, such as regular physical activity and a healthy diet, and other pharmacological measures, including oral antidiabetic drugs [4]. In addition, therapeutic alternatives of natural origin, such as plants, have been sought out due to their containing of bioactive compounds with pharmacological properties that intervene in antioxidant action or their mechanisms of action that regulate glucose, among others properties [5,6]. Plants are considered important in traditional medicine and human nutrition, among which edible fabaceous plants (legumes) stand out, such as soybeans, beans, peas, broad beans, lentils, peanuts, and chickpeas. It has been suggested that these fabaceous plants can help prevent or reduce the complications of diabetes treatment and may even have similar effects to insulin [7]. Compounds such as alkaloids, phenolics, carotenoids, flavonoids, lectins, tannins, trypsin, glycosides, coumarins, and saponins have been identified in fabaceous plants as those responsible for some biological activities, i.e., phenolic acids, flavonoids, and isoflavonoids are classes of bioactive compounds found in plants that are known for their antioxidant, anti-inflammatory, and potential health-promoting effects [8,9].

Phenolic compounds are secondary metabolites that provide plants with essential properties such as color, flavor, and resistance to stress. Beyond their organoleptic properties, these compounds are critical to the human diet because they contain antioxidant properties which prevent cell damage [10,11], meaning they can help confront damage caused by free radicals in the body [12]. A comprehensive view of the benefits of polyphenols with antidiabetic effects present in common fabaceous plants can motivate clinical trials, attracting multiple disciplines for further investigation. Given the growing concern among individuals with DM about the potential side effects of synthetic pharmaceutical agents, the research community is increasingly looking for natural products as a safer alternative. This study aims to analyze and synthesize the most recent scientific evidence on the phenolic compounds present in commonly consumed fabaceous plants, evaluate their potential as antioxidants and in glucose modulation, understand their role in preventing and managing diabetes, and ways to incorporate fabaceous plants as dietary strategies for diabetes.

## 2. Diabetes Mellitus Generalities

According to the International Diabetes Federation, in 2021, the number of people living with diabetes worldwide was 537 million, of whom more than 95% had type 2 diabetes mellitus (T2DM) [1,13]. DM is diagnosed in patients who meet one of the following criteria: glycohemoglobin A1C (≥48 mmol/mol), fasting plasma glucose (FPG) value (≥126 mg/dL), 2 h glucose value (≥200 mg/dL) during a 75 g oral glucose tolerance test (OGTT), or random glucose value (≥200 mg/dL) [14].

Beta cells, one of the cells of the pancreatic islet, synthesize and secret insulin, regulating blood glucose levels [15]. After a meal, glucose is transported in the blood and enters the beta cell through the glucose transporter Glut2, glucose is then metabolized through glycolysis, increasing the intracellular ATP/ADP ratio, which in turn leads to the closure of the ATP-dependent potassium (KATP) channels in the plasma membrane. This causes membrane depolarization and opening of voltage-gated calcium channels, allowing Ca2+ to enter the cell, triggering insulin exocytosis, leading to the release of insulin into the bloodstream [16]. Impaired pancreatic beta cell function has been identified as a major factor contributing to the onset of T2DM. The mechanisms leading to beta cell failure in type 1 diabetes are classified into three categories: (a) reduction in beta cell number, (b) functional exhaustion of beta cells, and (c) loss of their cellular identity [17,18]. In type 2 diabetes, insulin resistance is the primary cause, while impaired insulin secretion results from inflammation and hyperglycemia. Hyperglycemia is a common trigger in both forms of diabetes as it causes oxidative stress [19], inflammation, cytokine secretion, cellular exhaustion, and apoptosis, contributing to beta cell damage and disease complications [20].

Based on the above information, antidiabetic agents target different steps in the signaling and functioning of insulin, such as the activation of insulin receptors (insulin analogs) and downstream signaling in multiple sensitive tissues [4], insulin sensitizers (sulfonylureas, biguadines) increasing glucose uptake in tissues of the whole body [21,22], the opening of voltage-dependent calcium channels (meglitinides) [23], and inhibitors of SGLT-2 [24]. Additionally, GLP-1 analogs represent a significant advancement in the treatment of type 2 diabetes by improving glycemic control, promoting weight loss, and having a low risk of hypoglycemia [25].

The treatment of diabetes mellitus is based on the diagnosis of the type of diabetes mellitus, the available treatment regimens, lifestyle changes (diet and exercise), oral hypoglycemic drugs, such as biguanides, sulfonylureas, meglitinides, thiazolidinediones, gliptins, α-glucosidase and sodium-glucose cotransporter inhibitors, and, finally, insulin. An alternative antidiabetic medication should, as much as possible, be capable of preventing the onset and progression of T2DM and should stop the loss of beta cells and/or promote the restoration of beta cell mass independently of reducing hyperglycemia and ameliorating glucotoxicity and oxidative stress in pancreatic islets. Plants and their fruit could be a good alternative with few or no adverse effects in the treatment of diabetes [26].

For centuries, plants have been used to treat diseases, being the precursors of modern medicine. The use of plants was intended to reduce the discomfort caused by diseases, ensuring that the plant worked through trial and error. Based on cultural beliefs and experiences transmitted from generation to generation, different parts of plants, such as the seeds, flowers, leaves, roots, bark, fruits, and stems, were used without knowing the active ingredient or mechanism [27,28]. Plant extracts are used as supplements for treating diseases, becoming a natural therapeutic alternative with almost no side effects because they come from commonly consumed plants such as fabaceous ones [29,30].

## 3. Importance of Secondary Metabolites in Fabaceous Plants

Fabaceae are part of a family of nitrogen-fixing plants comprising 770 genera and approximately 19,500 species, which help to grow other plants in infertile or nutrient-poor soils. The plants of these species are climbers (annuals), herbs, aquatic plants, woody lianas, trees, shrubs, and subshrubs [8]. This family includes the fabaceous plant, commonly known as legumes. They are characterized by producing pods containing seeds (one to twelve) that vary in color, size, and shape depending on the type of plant. Many fabaceous plants are used for human consumption and/or oil extraction, as well as for animal feed, the most common being soybeans, beans, peas, broad bean, lentils, peanuts, and chickpeas due to the quality of their nutrients (complex carbohydrates, unsaturated fats, proteins, amino acids, vitamins, and minerals) and their low cost [31,32]. Fabaceous plants also contain secondary metabolites (SM) that act in the plants as a chemical defense against insects or predators and attract pollinators; on the other hand, when they are consumed, they present biological activities with benefits to health [28,33,34,35].

Secondary metabolites are chemical compounds produced by plants, fungi, and other microorganisms which are essential for interacting with the environment; the SM has a molecular mass of <3000 kDa, and they are distributed throughout the plant [34,36]. The stress (biotic or abiotic) to which a plant is subjected influences the production of SM; this can be caused by herbivores, pathogens, salinity, solar radiation, extreme temperatures, drought, and lack of nutrients (Figure 1). The SM, being a response to stress, turn into defenders of the plant, thus achieving its adaptation to the environment and its survival [34,37]. Plants, through their SM, can attract pollinating insects, as well as nitrogen-fixing bacteria, forming nodules of different sizes and characteristics in their roots [38]. SM can be classified into four main groups: phenolic compounds, nitrogen compounds, sulfur compounds, and terpenes. Phenolic compounds are distinguished by their ability to defend plants, functioning as antimicrobials and herbivore repellents. They also contribute to the protection against oxidative damage [36].

## 4. Methodology

The search for information was performed following the PRISMA (Preferred Reporting Items for Systematic Review and Meta-Analysis) on Google Scholar, PubMed, Science Direct, and Scopus. Of all the articles found, those published within the previous 5 years were considered, with a maximum limit of 10 years for the plants with scarce information on them (less than five articles), ensuring the inclusion of recent and relevant advances, reflecting the area’s current state. The keywords used for the search of articles were fabaceous (in general and for each of the species presented, it was also searched as legumes), “phenolic compounds”, “antidiabetic phenolic compounds fabaceous”, “extracts antidiabetic fabaceous”, “antioxidant activity fabaceous”, and each of the species names. From the selected articles, the general characteristics of fabaceous compounds, the phenolic compounds content, and the most outstanding compounds were considered, in addition to the results showing the hypoglycemic effect.

The inclusion and exclusion of articles in this review focused on selecting studies that addressed key aspects of diabetes mellitus, such as its description, development, and complications, as well as the production of secondary metabolites, especially phenolic compounds in fabaceous plants. Research that measured the antidiabetic potential of these phenolic compounds in commonly consumed species, such as soybeans, beans, peas, broad beans, lentils, and chickpeas, was included. However, articles that did not use the Fabaceae seed or its coating were excluded since these parts are the most relevant to studying the impact of phenolic compounds on diabetes management, ensuring that the review focused on the most pertinent and specific studies. Figure 2 shows the process of searching for information. First, we explored Google Scholar to determine whether the required information was available. Then, we used the most well-known databases and journals with a publication trend, precise information, and inclusion and exclusion criteria. In the record screening, a review of the titles and abstracts of the articles was carried out to evaluate their compliance with the established inclusion or exclusion criteria.

## 5. Antioxidant Activity of Total Phenolic Compounds from Fabaceous Plants

The main characteristics of phenolic compounds are an aromatic ring and a hydroxyl group; within these, we can find catechol, derivatives of hydroxybenzoic acid, condensed tannins, flavonoids, stilbenes, and lignans. Phenolic compounds, being plant protectors, can function as toxins for herbivores or alter the growth or physiological processes of insects due to oxidation to toxic metabolites [34].

On the other hand, phenolic compounds also give color to the products extracted from red fruits (i.e., juices and wines), contributing to the aroma and enzymatic browning of the fruits [39]. Furthermore, phenolic compounds from vegetables and fruits, legumes, and grains included in the human diet are associated with health benefits [11,39].

Table 1 shows that the content of phenolic compounds in legumes varies significantly depending on the type of legume, the specific variety, and the region where they are grown. These differences show how factors such as the genetic variety of the legume and the growing conditions specific to each geographical region directly influence the accumulation of phenolic compounds, which are essential for their antioxidant properties and health benefits. For example, the highest concentration of these compounds in soybeans is reported in wild soybeans, 41.53 ± 1.25 mg GAE/g, compared to cultivated soybeans, whose concentration is lower (approximately 12.5 mg GAE/g). On the other hand, the color of the different varieties of bean seeds is also reflected in the values of phenolic compounds, such as in black, velvet, red, and white, where there is a range of 197–1328 mg GAE/g [40,41]. Something similar happens with pea varieties, although the concentration is lower than in beans, ranging from 0.12 to 2.66 mg GAE/g, a considerably wide range. The phenolic compound profile in the fabaceous plants of Table 1 is unique and presents an extensive range due to variety (genetic diversity), environment, stage of maturity, growing conditions, and cultivation methods [34].

Moreover, the amount of phenolic compounds in fabaceous plants has been shown to correlate with antioxidant capacity and health benefits (Table 2). For example, in the case of soybeans, flavonoids predominate and have an associated relationship with biological activity, having a high antioxidant capacity of around 80% of the inhibition of free radicals through ABTS and DPPH reported [43]. In the case of beans, the presence of anthocyanins, which are responsible for giving red, blue, and purple colors, also influences the inhibitory activity of free radicals [47]. Although flavonoids are also reported in peas, they contain free phenolics that are more available to exert antioxidant activity. Fabaceous foods like beans and lentils have more phenolic compounds, influencing their antioxidant activity. However, this activity does not reflect much variation based on the color of the seed but rather on the profile of the compounds (Table 2).

Phenolic compound extraction and processing as phytotherapeutic practice has the potential to be effective therapeutic or preventative agents against different diseases [11,39], neutralizing free radicals and protecting cells from oxidative damage. This is known as antioxidant capacity, one of the most studied biological activities due to its relationship with chronic diseases. For example, in DM, an increase in free radicals affects the mechanism of insulin action, causing damage to pancreatic beta cells.

Phenolic compounds, as part of bioactive compounds, have been shown to significantly inhibit enzymes such as lipases, α-amylase, α-glucosidase, and β-glucosidase in vitro [69]. By inhibiting α-glucosidase, these compounds reduce glucose digestion and absorption in the intestine, which is crucial for managing type 2 diabetes as they help regulate blood sugar levels after meals. The activity of phenolic compounds underlines the importance of bioactive compounds in the diet, highlighting their potential to contribute to the prevention and control of metabolic diseases [69,70]. The bioactive compounds contained in fabaceous plants reduce the risk of suffering from T2DM [33,34]. In vitro antidiabetic studies of fabaceous plants have been evaluated through various methods, including glucose uptake assay and the inhibition of α-glucosidase, α-amylase, and DPP-IV (Dipeptidyl peptidase IV) [32].

## 6. In Vitro and In Vivo Antidiabetic Studies of Edible Fabaceous Plants

### 6.1. Soybean

Soybean (*Glycine max*) is one of the most edible fabaceous plants with the highest protein (40%) content; the wild soybean tends to have 10% more protein and 10% less oil than black soybean [71]. In addition to its nutritional composition, carbohydrates (33%), unsaturated fatty acids (20%), fiber, vitamins, minerals, and other bioactive compounds such as polyphenols, flavonoids, isoflavone, and glycosides are present [45,72,73]. The combination of phenolic compounds such as anthocyanins, proanthocyanidins, chlorophyll, and other pigments defines the characteristic color of their seed coat [42,74].

Identifying and analyzing the α-glucosidase enzyme inhibitors derived from black soybeans resulted in the bioactive compounds present in this fabaceous plant achieving inhibition of the enzyme; this lead to the regulation of glucose levels because the digestion of carbohydrates was reduced, which suggests a potential use for this inhibitor in the management of T2DM. Soy isoflavones (daidzin, glycitin, genistin, malonyldaidzin, malonylgenistin, genistein, and daidzein) inhibitors of α-glucosidase were identified, resulting in daidzein (IC_50_ 15.7 ± 0.3 μmol/L) and genistein (IC_50_ 3.2 ± 1.2 μmol/L) showing an inhibitory activity superior to that of acarbose (IC_50_ 632.5 ± 70.0 μmol/L). The structure–activity relationship indicated that isoflavone aglycones without glucosylation have higher inhibitory activity. Hydrophobic interactions and hydrogen bonds were the main forces involved in the interaction between isoflavones and α-glucosidase, suggesting that black soybean, through its isoflavones, has significant antidiabetic potential [72]. In another investigation conducted with the same animal model of T2DM, *Glycine max* fermented flour was used in a dose of 18.050 mg/kgBW, reducing blood glucose as the positive control group (administration of treatment). One of the hypoglycemic mechanisms is that isoflavone compounds can be transformed into aglycones (genistein, glycitein, and daidzein), which help reduce blood glucose. In addition, the fermentation process can transform the aglycone to produce compounds with greater biological activity, such as 6,7,4’ trihydroxy isoflavone, which has better antioxidant activity than daidzein and genistein. It also acts on free radicals caused by hyperglycemia. Isoflavones lower blood glucose by activating the peroxisome proliferator-activated receptor and protecting cells from cytokine pre-inflammation, fat-induced damage, and apoptosis [75].

Son et al. investigated the antidiabetic effects of *Glycine soja* extract in type 2 diabetic mice and insulin-resistant human hepatocytes for 6 weeks. Different concentrations were evaluated, with the group receiving the highest dose (300 mg/kg/day) achieving the lowest blood glucose levels (331.3 ± 78.6 mg/dL). Significant effects showed properties that improved insulin sensitivity by increasing adiponectin, and blood glucose and glycated hemoglobin levels were reduced, especially at doses higher than 150 mg/kg/day [71]. The hypoglycemic effect of fermented mulberry with soy has also been evaluated. This mixture for diabetic mice (type 2) reduced blood glucose levels, and improvements were observed in pancreatic function and insulin sensitivity. This indicates that the combination of mulberry and soy (1:5) enhances the beneficial effects at a concentration of 2.26 g/kg/day since the diabetic group reduced food consumption, improved glucose tolerance, and optimized the blood lipid profile, thanks to its antioxidant capacity [76].

### 6.2. Beans

Beans (*Phaseolus vulgaris*) are one of the most widely consumed fabaceous plant, with black beans being one of the main ones included in the diets of Latin America and Africa. This fabaceous plant stands out due to its rich nutritional composition of protein (17.9–31.1%), carbohydrates (25–60%), lipids (0.55–2.1%), fiber (4–20%), vitamins, and minerals. Despite being rich in carbohydrates, they have a low glycemic index, so they release their energy slowly, helping to keep blood sugar levels stable [77]. This fabaceous plant is rich in nutrients such as carbohydrates, proteins, fiber, and bioactive compounds (phenols, alkaloids, phytosterols, coumarins, and saponins). It has also been reported that they contain polysaccharides, peptides, and polyphenols. Bean seeds (*Phaseolus vulgaris*) are used as traditional medicine in some parts of China [78]. Polyphenol extracts in mung beans (a common compound in plants) have a hypoglycemic effect. By consuming foods with a low glycemic index and high dietary fiber content, the blood glucose is reduced, facilitating the elimination of glucose through the glucose transporter (GLUT4) [79,80].

A study evaluating the inhibitory effects of polyphenol-rich extracts (from six bean varieties) on α-amylase and α-glucosidase enzymes found that the *Sanghellato* variety had the highest amount of phenolic compounds. In carbohydrate digestion, polyphenolic extracts showed an inhibition of the α-amylase enzyme as well as inhibiting the α-glucosidase enzyme. The *Screziato Impalato* variety presented the best inhibition of α-amylase with an IC_50_ value of 69.02 µg/mL. For α-glucosidase, there was a more significant inhibition, presenting an IC_50_ of 90.40 µg/mL for the variety *Cannellino Rosso*. It is reported that tannins and proanthocyanidins (phenolic compounds) are related to the inhibition of both enzymes, suggesting a synergistic interaction between phenolic compounds and the inhibitory effect [81].

In a mouse model with T2DM, the hypoglycemic effect of beans was evaluated, where a significant reduction in blood glucose, cholesterol, and lipid levels was observed due to the influence on metabolic pathways related to insulin sensitivity and the regulation of lipid and carbohydrate metabolism [82]. The identified metabolites present significant potential for managing T2DM, related to the enrichment of soybean sprouts with γ-aminobutyric acid (GABA). This compound has been studied for its ability to improve metabolism and regulate blood glucose levels. It has been suggested that the enrichment of certain foods, such as soybean sprouts, with GABA could have beneficial effects on metabolic health and the control of type 2 diabetes. In addition, a decrease in urea and creatinine levels was observed, indicating that several metabolic parameters are improved in diabetic mice [82]. On the other hand, the effect of polyphenol extracts from germinated mung beans was evaluated in T2DM mice, obtaining significant improvements in blood glucose levels and the reduction in systemic inflammation associated with T2DM. Polyphenolic extracts reduce fasting glucose levels, improve glucose tolerance, and decrease insulin resistance, providing a better effect in the high concentration group (150 mg/kg) of the extract and improving lipid levels and liver enzymes. Along with these results, a balance was observed in the mice’s intestinal microflora related to insulin sensitivity [83]. In another study, they evaluated the effect of bean-rice and rice-without-bean intake in adults with T2DM with consumption at different times; a significant difference was obtained in the groups that consumed bean-rice compared to those that did not. Three varieties of beans were evaluated; the groups that ingested pinto and black beans showed lower glycemic values than the red beans and the control (rice only). This study suggests that including beans in meals can reduce the glycemic response in patients with T2DM, offering a non-pharmacological dietary management alternative. In addition, promoting these traditional foods can improve dietary adherence and quality of life in minority and immigrant populations with diabetes [84]. Knowing the effect of the compounds present in beans is important, as few studies focus on the substances. Although the main phytonutrients studied as phytotherapeutics are secondary metabolites, the study of the various components of foods should be expanded.

### 6.3. Pea

The pea (*Pisum sativum*) is a bean of European origin [65] that is very adaptable to cultivation. Peas have a high nutritional value due to their composition of dietary fiber (11.34–16.13%), lipids (0.57–3.52%), and protein (19.75–26.48%), in addition to trace elements, phenolic compounds, and their glycemic index being less than 60 (considered medium or low). Flavonoids are the main polyphenols in peas so the content of their bioactive compounds can be used for functional foods or other products [50,85].

Di Stefano et al. investigated how bioprocessing (germination and fermentation) of fabaceous plants influences the inhibitory activities of DPP-IV and α-glucosidase. The findings indicated that bioprocessing facilitates the digestion and absorption of bioactive compounds, and that yellow pea extract can inhibit two critical enzymes in blood glucose regulation: DPP-IV and α-glucosidase. The inhibition of DPP-IV was approximately 55.1 ± 1.5 milliequivalents of Diprotin A, while the inhibition of α-glucosidase was 56.5 ± 5.2 milliequivalents of acarbose, suggesting that processed fabaceous plants could function as health supplements [85]. In another study, the inhibitory activity of the enzyme α-glucosidase from pea (*Cajanus cajan*) extract was tested. This extract containing saponins, flavonoids, phenolics, and tannins, among other compounds, caused a delay in the breakdown of carbohydrates, leading to a decrease in glucose absorption into the bloodstream and thus lowering postprandial hyperglycemia. The enzyme α-glucosidase results in an IC_50_ value of 69.67 ppm, a high level which indicates its potential as an antidiabetic drug [86]. The consumption of peas has presented advantages for potentially improving diseases associated with insulin resistance. Peas with a low glycemic index (22) can help maintain blood sugar levels, which can trigger an improvement in insulin sensitivity [51].

### 6.4. Broad Beans

*Vicia faba*, known as the broad beans, is a fabaceous plant rich in nutrients such as protein (present in high amounts; 29%), carbohydrates (56–68%), lipids (2.30–3.91%), fiber, vitamins, and phenolic compounds in the seed. Due to the root nodules, the bean plant has a greater capacity to fix nitrogen than other bean plants. The consumption of broad bean flowers benefits health due to their polyphenol content, which reduces the risk of cancer, cardiovascular diseases, and diabetes due to their ability to counteract free radicals [87,88]. The broad bean has a low glycemic index and little fat; this seed is a health promoter because it has anti-cancer, anti-diabetic, anti-obesity, and cardioprotective effects. The primary polyphenols in broad beans are tannins; broad bean seeds usually have many phenolic compounds, almost twice as many as other fabaceous plants [89]. Mejri et al. report different percentages in extracting broad bean’s yield of phenolic compounds because different solvents were used to obtain them. Methanol presented the highest extraction with 25.8%, ethanol with 17.5%, butanol and ethyl acetate presented the lowest extraction percentage, at 11.3% and 0.81%, respectively. The methanolic extract also exhibited the highest total phenolic, flavonoid, and tannin content and showed antidiabetic effects in alloxan-induced diabetic mice. It was found that blood glucose concentration and deterioration of pancreatic β-cells produced by alloxan were reverted after the administration of BBP extract, apparently through its antioxidant properties [53]. Sharma and Giri reported an α-amylase inhibitory concentration IC_50_ of 264.69 μg/mL, thus providing data on the properties of natural antidiabetic agents of the broad bean; this result was better than the standard (acarbose IC_50_ of 52.76 μg/mL) [9]. When extracts are obtained with different solvents, there may be differences in the potential for inhibition of α-amylase, as shown in the study of the broad bean where the best concentration was 3-5 mg/mL with the acetone and methanol extracts having an IC_50_ value of 2.94 mg/mL, which could lead to a lower postprandial glucose level [90]. Broad beans are a rich source of polyphenolic compounds acting as antioxidants scavenging free radicals, helping in treating diabetes and, with the rejuvenation of beta cells of the pancreas, broad beans are ideal for consumption by diabetics as they inhibit α-glucosidase, thus delaying the absorption of carbohydrates [88].

### 6.5. Lentils

The lentil (*Lens culinaris*) contains many macro and micronutrients: carbohydrates (40–50%), protein (20–30%), fiber, vitamins, and minerals, highlighting that the green and gray seeds are preventative and, in many cases, have a health benefit [57,58,66]. Among the benefits of consuming lentils is their insoluble fiber content, as a source of prebiotics and prebiotic carbohydrates, stimulating the microbial flora, benefiting health, and preventing intestinal diseases. Lentils (sprouted) have been reported to improve blood glucose metabolism as well as decrease lipoproteins and lipids in diabetic patients. Compared to other fabaceous plants, lentils contain a high content of phenolic compounds, with a high concentration of phenolic acids, flavonoids, and condensed tannins. The polyphenols in lentils reduce the glycemic index, making them suitable for a healthy diet [57]. The consumption of lentils is recommended to prevent or control diabetes since it has been shown to improve the metabolism of lipids and lipoproteins, as well as help with blood glucose; the above is possible due to the fiber content, in addition to the flavonoids contained in them [58].

Magro et al. found phenolic compounds such as ferulic acid and quercetin in fermented lentils; vanillic acid and 3,4-dihydroxybenzoic acid were also detected (in fermented and non-fermented lentils). Lentils fermented with *Aspergillus niger* showed greater inhibition (91%; 48 h of fermentation) of α-glucosidase than those fermented with *Aspergillus oryzae* and non-fermented ones. For the inhibition of α-amylase, lentils were fermented with *A. oryzae* at different fermentation times (75%; 24 h, 73%; 48 h and 71%; 0 h); α-amylase is an enzyme that breaks down starches into simple sugars, and its activity may influence blood glucose control [66].

In an experimental study with diabetic male and female mice, methanolic extracts of fermented lentil seeds were administered for 6–8 weeks. Aqueous methanolic extracts of fermented lentils reduced blood glucose in diabetic mice; the reduction was significantly greater with the 400 mg/kg extract (169.92 ± 1.62 mg/dL) than with the 200 mg/kg extract (180.83 ± 2.858 mg/dL) and 100 mg/kg (190.83 ± 1.80 mg/dL). All three doses showed a statistically significant difference from the diabetic control group with blood glucose concentration (252.17 ± 3.84 mg/dL). The group treated with glibenclamide showed a hypoglycemic effect due to its direct impact on the mechanics of insulin release from beta cells and increasing glucose tolerance [91].

### 6.6. Chickpea

The chickpea is a fabaceous plant with a rich nutritional composition containing 17 to 22% of proteins, 18 to 22% of dietary fiber, and a higher lipid content of up to 7% [61]. Chickpea bioactive compounds, such as phenolics, help inhibit the hydrolysis of carbohydrates and some lipids, decreasing the risk of developing T2DM. Polyphenols also have a mechanism of action to inactivate the DPP-IV enzyme by 70-90% [92]. Seed chickpeas are other well-known and consumed fabaceous plants, but little research has been performed on their phenolic compounds with antidiabetic potential. Chickpea (*Cicer arietinum*) is characterized by its high protein content but also by the phenolic compounds it has, such as isoflavones (153 to 340 mg/100 g) [44].

Based on different experimental studies, the Table 3 summarizes the antidiabetic effects of various fabaceous, such as soybean, bean, pea, broad bean, lentil, and chickpea. Key mechanisms are highlighted, such as the inhibition of enzymes related to carbohydrate metabolism (α-glucosidase and DPP-IV), antioxidant activity, reduction in glucose, lipid, and glycosylated hemoglobin levels, as well as improvement in insulin sensitivity and decreased inflammation. This summary demonstrates fabaceous therapeutic potential in controlling and managing diabetes.

## 7. Perspectives

Natural products have been used as medicine throughout history and are still the basis for the development of drugs. Current research not only seeks new molecules but also focuses on the development of phytotherapeutics to prevent or treat diseases or find evidence to support the biological activity of plants. The design and development of phytotherapeutics must be standardized, and the chemical compounds associated with the medicinal properties present in plants must be identified. Phytotherapeutics or nutraceuticals can be formulated from bioactive compounds such as phenolics (Figure 3). Identifying phenolic compounds with antidiabetic potential can produce natural products in combination with pharmacological products that help prevent and/or treat diabetes mellitus, assisting patients with diabetes in the future. The analysis of bioactive molecules in plants drives the development of natural products, providing a new option for treating different diseases.

Generally, plant consumption is assumed to be safe; however, it is important to consider that they contain chemical compounds that, in some cases, can cause oxidative damage in the body. Despite this, current evidence widely supports the safety of phytochemicals for treating DM. Improving the stability and bioavailability of phytotherapeutics is essential to ensure adequate absorption and maximize their efficacy as a treatment, offering an effective alternative with fewer side effects compared to conventional drugs used in the management of DM [93]. Phenolic compounds may have properties that can affect human health because they could act as endocrine disruptors, interfering with hormonal functions and influencing cellular processes such as energy production and generating reactive oxygen species [94].

Phenolic compounds exert their beneficial effects on health through a series of biochemical mechanisms that intervene at both the cellular and molecular level (Figure 4). These compounds, thanks to their chemical structure that includes a hydroxyl group (-OH), have the ability to donate electrons to free radicals or ROS species, neutralizing them and preventing cellular damage that can result from oxidative stress. This process protects cells from the damage that ROS can cause to key components, such as lipids, proteins and DNA. Regarding inflammation, phenolic compounds also modulate this response by reducing the release of proinflammatory cytokines such as TNF-α, IL-1β and IL-6, molecules that amplify and perpetuate inflammatory processes. In addition, many of these compounds have the ability to inhibit the activation of NF-κB (nuclear factor kappa B), a key protein in the regulation of chronic inflammation. Inhibition of NF-κB disrupts signals that promote persistent inflammation, which is crucial to preventing the development of various chronic diseases [93,95].

Although phenolic compounds may have certain adverse effects, they also have beneficial properties, such as their antioxidant capacity, which helps neutralize free radicals and protect cells from oxidative damage. In addition, some phenolic compounds are widely studied for their possible applications in the prevention of chronic diseases, such as cardiovascular diseases and certain types of cancer, thanks to their protective effects on the body when used in a controlled manner.

## 8. Conclusions

Fabaceous plants are accessible to virtually everyone and are essential for a healthy diet due to their nutritional profile. For some people, fabaceous plants are their primary source of protein as they avoid animal protein. The importance of fabaceous plants is highlighted by their bioactive compounds, such as phenols, which function as antioxidants and anti-inflammatory agents and play an essential role in health, stabilizing free radicals and preventing cell damage. Oxidative stress caused by free radicals in cells is associated with diabetes mellitus, which contributes to complications in different organs. The action of phenolic compounds helps mitigate damage and reduce risks due to their antioxidant and anti-inflammatory properties, benefiting patients with different pathologies.

Phenolic compounds have been evaluated in vitro and in vivo, where favorable results have been observed not only for their antioxidant capacity but also for their hypoglycemic effect, positively impacting the health of patients with diabetes mellitus; these characteristics make fabaceous plants of great interest with the potential to be used in future nutraceutical applications due to their efficacy.

The results of different studies summarized here support the role of fabaceous plants as adjuvants in the control of diabetes and show the importance of integrating the knowledge of traditional medicine with pharmacological medicine to improve the care of patients suffering from this condition. To integrate these herbal remedies into modern medicine, further research on phenolic compound type, doses, efficacy, and safety in humans is needed.

## Figures and Tables

**Figure 1 pharmaceuticals-18-00069-f001:**
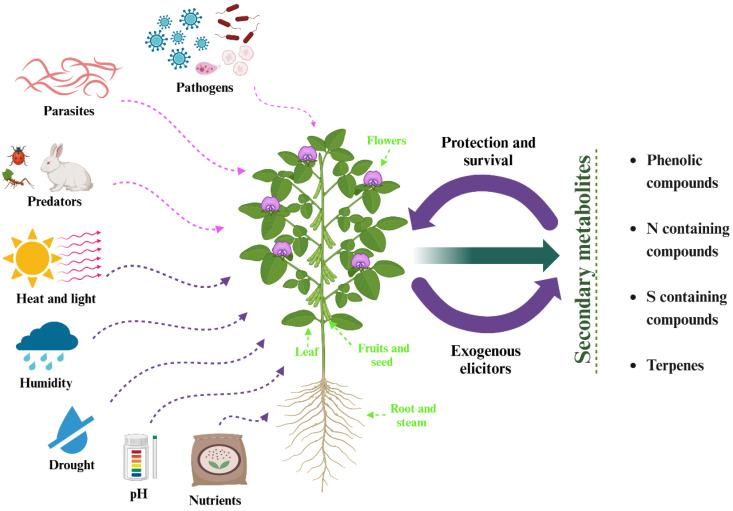
Secondary metabolite production in plants (N = nitrogen, S = sulfur) (created by BioRender).

**Figure 2 pharmaceuticals-18-00069-f002:**
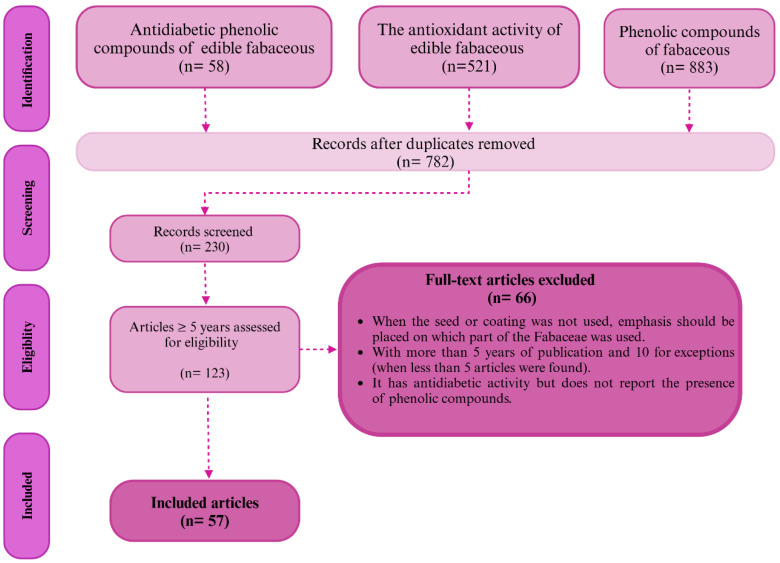
General diagram for the identification and selection of articles (created by BioRender).

**Figure 3 pharmaceuticals-18-00069-f003:**
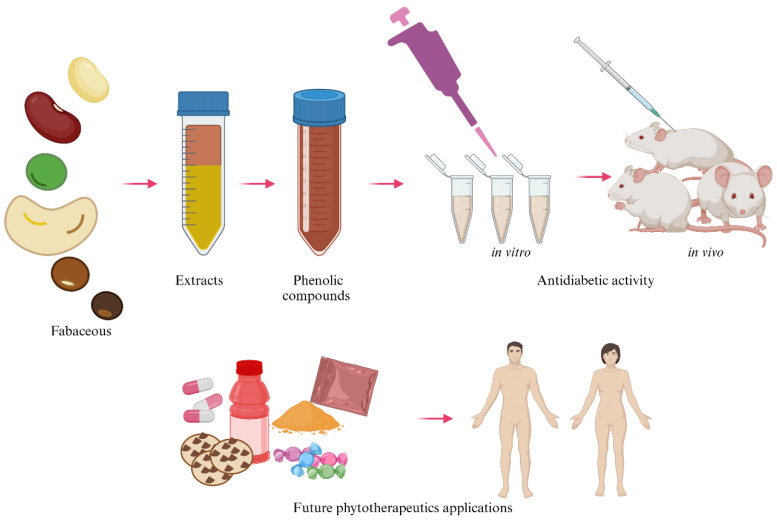
The process of obtaining and developing phytotherapeutics from fabaceous plants (created by BioRender).

**Figure 4 pharmaceuticals-18-00069-f004:**
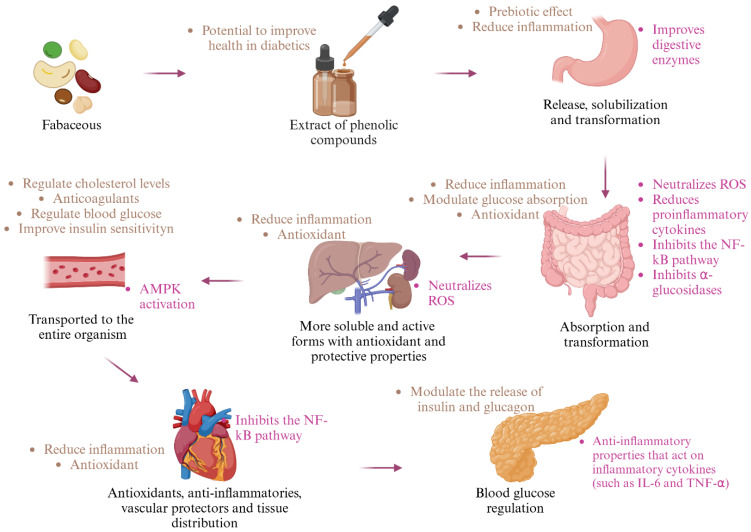
Potential mechanisms of action of phenolic compounds [93,95] (created by BioRender).

**Table 1 pharmaceuticals-18-00069-t001:** Phenolic compound content in fabaceous seed.

Fabaceous	Variety	Total Phenolic Content	Reference
**Soybean**	Black soybean	830.66 ± 5.46 mg GAE/kg	[42]
	Wild soybean	41.53 ± 1.25 mg GAE/g	[43]
	Cultivated soybean	12.5 mg GAE/g	
	Glycine max	2.04–6.10 mg GAE/g	[44]
	Local soy (China)	43.87± 3.22 mg GAE/100 g	[45]
**Bean**	*Different varieties*	0.57–10.34 mg de GAE/g	
	*Black bean*	197.23 ± 0.02 mg GAE/g	[40]
	*Red kidney bean*	1174.32 ± 103.11 mg GAE/g	[41]
	*Velvet beans*	12.19–37.30 mg GAE/g	[46]
	*White kidney bean*	1328.30 ± 156.63 mg GAE/g	[47]
	*Vigna unguiculata*	337.6 ± 14.6 mg GAE/100 g	
	*Vigna mungo*	547.2 ± 14.1 mg GAE/100 g	[48]
	*Proteus vulgaris*	440.2 ± 12.7 mg GAE/100 g	
**Pea**	Different varieties	12.6–128.6 mg GAE/100 g	[49]
	Different varieties	0.66–2.66 mg GAE/g	[50]
	Yellow pea	0.85–1.14 mg GAE/g	[51]
	Green pea	0.65–0.99 mg GAE/g	
**Broad bean**	Commercial (Tunisia)	115.21 mg GAE/g	[52]
	Commercial (Italy)	2.06 mg CAE/g	[53]
	Cultivated (Australian)	258–570 mg GAE/100 g	[54]
	Cultivated (Korea)	3.61 ± 0.11 mg GAE/g	[55]
	Cultivated (Tunisia)	1.122–1.225 mg GAE/g	[56]
**Lentil**	Commercial	26 mg GAE/100 g	[57]
	Different varieties	4.6–70 mg GAE/g	[58]
	Cultivated (Argelia)	49.65–59.12 mg GAE/g	[59]
	Black lentil	0.84 ± 0.03 mg GAE/g	[60]
	Green lentil	0.96 ± 0.07 mg GAE/g	
	Brown lentil	0.79 ± 0.02 mg GAE/g	
	Red lentil	0.81 ± 0.12 mg GAE/g	
**Chickpea**	Cicer reticulatum	8.02–10.84 mg GAE/g	[44]
	Different varieties	27.48–113.30 mg GAE/100 g	[61]

GAE: Gallic acid equivalent. CAE: Chlorogenic acid equivalents.

**Table 2 pharmaceuticals-18-00069-t002:** Antioxidant activity of phenolic compounds from fabaceous plants.

Fabaceous Plant	Main Associated Compounds	ABTS	DPPH	Reference
**Soybean** (*Glycine max*) 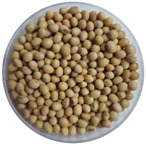	Flavonoids Anthocyanins Flavonoids (soybean; wild and cultivated) Flavonoids Flavonoids	92.74 ± 1.47 mmol eq. trolox/kg 84.28% inhibition 38.01% inhibition ----- EC_50_ 1.43 ± 0.12 mg/mL EC_50_ 1.64 ± 0.03 mg/mL	----- 82.58% inhibition 71.08% inhibition IC_50 =_ 58.33 ± 0.26 µg/mL EC_50_ 1.33 ± 0.01 mg/mL EC_50_ 1.29 ± 0.02 mg/mL	[42] [43] [9] [62]
**Bean** (*Phaseolus vulgaris*) 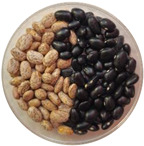	Phenolics (red and white *P. vulgaris*) Flavonoids Anthocyanins (*V. mungo*, *P. vulgaris*, and *V. angularis*) Flavonoids Phenolics	1/EC_50_ 0.0061 ± 0.0008 mL/mg 1/EC_50_ 0.0035 ± 0.0005 mL/mg 56.4 ± 1.5 µmol eq. trolox/g 51.8 ± 1.0 µmol eq. trolox/g 84.1 ± 0.8 µmol eq. trolox/g ----- 1190.32 ± 42.77 mg eq. trolox/g	----- 52.1 ± 4.8 µmol eq. trolox/g 53.7 ± 3.6 µmol eq. trolox/g 98.7 ± 1.6 µmol eq. trolox/g IC_50_ 69.74 ± 0.08 µg/mL -----	[47] [48] [9] [63]
**Pea** (*Pisum sativum*) 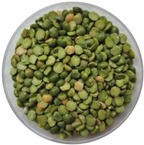	Flavonoids Flavonoids Free phenolic Bound phenolic Phenolics acids Flavonoids	----- ----- ----- 18.3 ± 1.2 mmol eq. trolox/kg	12.49 µmol eq. trolox/g IC_50_ 56.33 ± 0.13 µg/mL 2.3 µmol eq. trolox/g 0.5 µmol eq. trolox/g 13.7 ± 1.1 mmol eq. trolox/kg	[50] [9] [64] [65]
**Broad bean** (*Vicia faba*) 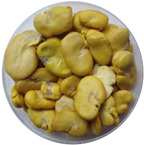	Flavonoid Condensed tannin Flavonoid Flavonoids	IC_50_ 610.61 ± 0.56 µg/mL ----- IC_50_ 2.35 ± 1.81 µg/mL	IC_50_ 157.94 ± 0.56 µg/mL IC_50_ 59.60 ± 0.24 µg/mL IC_50_ 74.71 ± 2.91 µg/mL	[52] [9] [53]
**Lentil** (*Lens culinaris*) 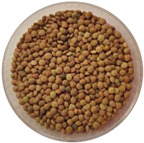	Flavonoids Condensed tannin Flavonoid Condensed tannin Proanthocyanin Anthocyanin (black, green, red, and brown) Flavonoids Phenolics acids Flavonoids (fermented seed; *A. oryzae* 24 h and *A. niger* 48 h)	EC_50_ 1.28 mg/mL 6.47 ± 0.81 mg eq. trolox/g 9.82 ± 0.71 mg eq. trolox/g 8.06 ± 0.47 mg eq. trolox/g 7.94 ± 0.12 mg eq. trolox/g ----- 6.81 µg eq. trolox/g 6.97 µg eq. trolox/g	EC_50_ 4.92 mg/mL 4.32 ± 0.08 mg eq. trolox/g 5.24 ± 0.02 mg eq. trolox/g 4.65 ± 0.02 mg eq. trolox/g 3.21 ± 0.08 mg eq. trolox/g IC_50_ 61.50 ± 0.16 µg/mL 7.48 µg eq. trolox/g 10.02 µg eq. trolox/g	[59] [60] [9] [66]
**Chickpea** (*Cicer arietinum*) 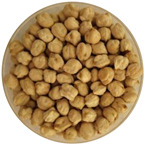	----- Flavonoids Isoflavonoids	25.06 mg/g de GAE 159–207 mmol eq. trolox/100 g	22.2 mg eq. trolox/g -----	[67] [68]

ABTS: 2,2’-azino-bis(3-ethylbenzothiazoline-6-sulfonic acid) and DPPH: 2,2-diphenyl-1-picrylhydrazyl; most used methods to evaluate antioxidant activity. Extracts or fractions of compounds with low IC_50_ and EC_50_ values are potent antioxidants. IC_50_: concentration corresponding to 50% inhibition. EC_50_ is the sample concentration required for a 50% decrease in DPPH concentration.

**Table 3 pharmaceuticals-18-00069-t003:** Impact of fabaceous on diabetes control.

Fabaceous	Experimental Approach	Effect	Reference
Soybean	Isoflavones in vitro Isoflavones in vitro Polyphenolic in vivo Polyphenolic in vivo	α-glucosidase inhibition Antioxidant activity Reducing blood glucose Reducing blood glucose and glycated hemoglobin Insulin sensitivity	[72] [75] [71] [76]
Bean	Tannins and proanthocyanidins in vitro Polyphenolic in vivo Polyphenolic in vivo Clinical study	α-glucosidase and α-glucosidase enzyme inhibition Reducing blood glucose, cholesterol, and lipid levels Improves blood glucose levels and reduces inflammation Lower glycemic values	[81] [82] [83] [84]
Pea	Polyphenolic in vitro Flavonoids and tannins in vitro	Inhibitory activities of DPP-IV and α-glucosidase Inhibitory activity of the enzyme α-glucosidase	[85] [86]
Broad beans	Flavonoids and tannins in vivo Polyphenolic in vitro	Blood glucose concentration and pancreatic beta cell deterioration were reversed Inhibitory activity of the enzyme α-glucosidase	[53] [9,90]
Lentil	Polyphenolic in vitro Polyphenolic in vivo	α-glucosidase and α-glucosidase enzyme inhibition Reducing blood glucose	[66] [91]
Chickpea	Polyphenolic in vitro	Inactivate the DPP-IV enzyme	[92]

## Data Availability

Data sharing does not apply to this article.

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
