# Peer review of "Polyphenolic Compounds in Fabaceous Plants with Antidiabetic Potential"

_pharmaceuticals, 2025, doi:10.3390/ph18010069_

Round 1
Reviewer 1 Report
Comments and Suggestions for Authors
This review explores a sensitive topic, i.e., the potential utility of natural products with antidiabetic potential in the improvement of diabetes management. The methodology should be better described, and the novelty of this review needs to be better substantiated.
Line 18- in the „Abstract”, the acronym „DM” should be placed after the first time „diabetes mellitus is used”.
Lines 72-74- Why were only papers published in the last five-ten years selected for this review?
The methodology is ambiguous. What inclusion and exclusion criteria were used? Is there a PICO/PICOS/SPIDER table available? Was the quality of sources assessed? Were both primary and secondary reports allowed?
It is difficult to understand how the selection of the relevant papers was conducted, because some sources are missing, although they were published in the last 5-10 years- for example, https://pubmed.ncbi.nlm.nih.gov/32708949/, or https://pmc.ncbi.nlm.nih.gov/articles/PMC9167359/. Such sources may be used at least for comparison reasons with the current review.
Section 3 seems like a part of the „Introduction”, since it contains generalities about diabetes mellitus. Usually, after the methodological chapter, follows the „Results” of the review (i.e., the retrieved reports).
The same observation can be formulated for chapters 4 and 5.
It looks like only chapter 6 includes the results of the review, as can be derived from the objective formulated in lines 65-67.
A table presenting the anti-diabetic properties of all products listed in section 6, focused on clinical and preclinical studies, would be helpful for the readers.
I could not find the reference (93) in teh body text, and it is important since there are two figures that look like they are made with bioRender.
Line 428- DM was already defined in the body test (line 37).
Author Response
This review explores a sensitive topic, i.e., the potential utility of natural products with antidiabetic potential in the improvement of diabetes management. The methodology should be better described, and the novelty of this review needs to be better substantiated.
Thank you for your comments. They were considered, and the changes were made.
Line 18- in the „Abstract”, the acronym„ DM” should be placed after the first time „diabetes mellitus is used”.
Thanks for your comment. The corrections have been made.
Lines 72-74- Why were only papers published in the last five-ten years selected for this review?
The 5-10-year period is ideal because it allows for incorporating the most recent and relevant developments, ensuring that the information reflects the field's most up-to-date and advanced state of the art. In addition, this approach minimizes the risk of including data that may have become obsolete or are no longer applicable in the current context (lines 74-75).
The methodology is ambiguous. What inclusion and exclusion criteria were used? Is there a PICO/PICOS/SPIDER table available? Was the quality of sources assessed? Were both primary and secondary reports allowed?
This process of inclusion and exclusion allowed us to focus the review on studies that contribute to understanding the role of phenolic compounds in fabaceous seeds about diabetes mellitus, avoiding research that was not aligned with the specific objectives of the review. This is marked in the figure, and thanks to your comment, it was added to the text (lines 82-89).
It is difficult to understand how the selection of the relevant papers was conducted, because some sources are missing, although they were published in the last 5-10 years- for example, https://pubmed.ncbi.nlm.nih.gov/32708949/, or https://pmc.ncbi.nlm.nih.gov/articles/PMC9167359/. Such sources may be used at least for comparison reasons with the current review.
An additional paragraph has been included detailing the article selection process. We have reviewed the suggested literature and incorporated that relevant to the paper's focus. We deeply appreciate your comments and suggestions.
Section 3 seems like a part of the „Introduction”, since it contains generalities about diabetes mellitus. Usually, after the methodological chapter, follows the „Results” of the review (i.e., the retrieved reports).
We appreciate your comments and suggestions. Appropriate adjustments have been made to the sections as indicated.
The same observation can be formulated for chapters 4 and 5.
We appreciate your comments and suggestions. Appropriate adjustments have been made to the sections as indicated.
It looks like only chapter 6 includes the results of the review, as can be derived from the objective formulated in lines 65-67.
We appreciate your comment. The objective has been adjusted accordingly concerning the scope of the review.
A table presenting the anti-diabetic properties of all products listed in section 6, focused on clinical and preclinical studies, would be helpful for the readers.
Thank you for your suggestion, which we consider very appropriate. For this reason, we have proceeded to prepare Table 3.
I could not find the reference (93) in the body text, and it is important since two figures look like they were made with bioRender.
Thank you very much for your comment. We have made the necessary corrections, and now each figure's citation is duly included.
Line 428- DM was already defined in the body test (line 37).
Thank you for the observation. The corrections have been made.

Reviewer 2 Report
Comments and Suggestions for Authors
In my opinion, the manuscript requires substantial revisions and a thorough rewrite to address several significant issues.
- Mechanisms of Antidiabetic Compounds:
The authors should provide an in-depth discussion of the possible mechanisms of action of antidiabetic compounds and secondary metabolites specific to legume plants. The current description of secondary metabolites is too general. For example, sulfur-containing metabolites mentioned are more typical of Brassica or Allium species (garlic, onion) rather than legumes. The focus must shift to secondary metabolites uniquely associated with legumes. - Figures:
- Figure 2: While visually appealing, it is not relevant to the manuscript's focus on legume secondary metabolites and should be revised to reflect this focus.
- Figure 3: This figure is unnecessary and can be removed.
- The statement, “Table 1 shows the variability of phenolic compound concentration depending on cultivation, type, region,” is overly general. Reporting the total amount of polyphenols is insufficient. In the current era of advanced analytical methods like HPLC and LC/MS, a more detailed presentation of the quality and quantity of active compounds, such as isoflavones, lignans, and phenolic acids, is essential.
- Table 2:
While antioxidant activity is important, it represents only one aspect of the potential antidiabetic activity of legume plants. The table should include a broader spectrum of bioactivities relevant to diabetes management. - Germinated Parts of Legumes:
The discussion focuses on seeds but neglects germinated parts of legumes. Germination significantly alters the composition of secondary metabolites, and this should be highlighted. Tables should include data on the bioactive compounds in germinated legumes. - Search Methodology:
The search methodology appears insufficiently robust, with notable omissions, such as the absence of information on certain plants like jicama roots. A well-planned systematic review protocol should be utilized, such as the OSF system, and its registration number provided. While not mandatory, this is crucial for transparency and allows editors and readers to verify the planning process. - Human Studies:
A major shortcoming of the paper is the lack of inclusion of human studies. There are numerous studies on the effects of legumes on diabetes management in patients. These must be incorporated for a comprehensive review. The conclusion suggesting future perspectives on this topic is misleading, as such studies are already being conducted. - Other Comments:
- Line 47: Additional mechanisms should be mentioned.
- Line 53: Rewrite this sentence for specificity—e.g., phenolic acids, flavonoids, and isoflavonoids should be explicitly mentioned.
- Line 108: The manuscript fails to mention mechanisms involving the latest antidiabetic drugs, such as GLP-1 analogs. This omission needs rectification.
- Line 208: The statement regarding the inhibition of enzymes by phenolic compounds needs proper referencing.
- Line 289: Clarify the impact of γ-aminobutyric acid (GABA) on diabetes management. Provide relevant studies and data to support its inclusion.
Author Response
Mechanisms of Antidiabetic Compounds:
The authors should provide an in-depth discussion of the possible mechanisms of action of antidiabetic compounds and secondary metabolites specific to legume plants. The current description of secondary metabolites is too general. For example, sulfur-containing metabolites mentioned are more typical of Brassica or Allium species (garlic, onion) rather than legumes. The focus must shift to secondary metabolites uniquely associated with legumes.
Thank you for your comments. They were considered, and the changes were made.
Figures:
Figure 2: While visually appealing, it is not relevant to the manuscript’s focus on legume secondary metabolites and should be revised to reflect this focus.
We appreciate your comment. We consider the image representative, as it clearly shows the location of the secondary metabolites and explains their formation. It also illustrates the production of phenolic compounds, which are the basis of the research.
Figure 3: This figure is unnecessary and can be removed.
We appreciate your comment; we have considered it and removed the image, as you suggested.
The statement, “Table 1 shows the variability of phenolic compound concentration depending on cultivation, type, region,” is overly general. Reporting the total amount of polyphenols is insufficient. In the current era of advanced analytical methods like HPLC and LC/MS, a more detailed presentation of the quality and quantity of active compounds, such as isoflavones, lignans, and phenolic acids, is essential.
Thank you for your feedback. We have corrected the table's description to make it more precise and understandable. We have generally described these compounds since not all sources or varieties include specific information on them. In this way, the information presented is more coherent and accessible, highlighting only the general concentrations and the differences between types of legumes, varieties, and growing regions.
Table 2:
While antioxidant activity is important, it represents only one aspect of the potential antidiabetic activity of legume plants. The table should include a broader spectrum of bioactivities relevant to diabetes management.
Thank you for your recommendation. Considering that other additional suggestions were received, it was decided to prepare a new table (Table 3) in which this information is more detailed.
Germinated Parts of Legumes:
The discussion focuses on seeds but neglects germinated parts of legumes. Germination significantly alters the composition of secondary metabolites, and this should be highlighted. Tables should include data on the bioactive compounds in germinated legumes.
In our analysis, we decided to focus exclusively on Fabaceae seeds and not consider germinated ones since our main objective was to evaluate their characteristics in their original state, such as their chemical composition, structure, and viability. Germination generates significant changes in these aspects, which could introduce variability and make comparing samples difficult. Furthermore, focusing on non-germinated seeds allowed us to maintain an approach consistent with the purpose of the study, aimed at analyzing the initial properties of the seeds before any transformation.
Search Methodology:
The search methodology appears insufficiently robust, with notable omissions, such as the absence of information on certain plants like jicama roots. A well-planned systematic review protocol should be utilized, such as the OSF system, and its registration number provided. While not mandatory, this is crucial for transparency and allows editors and readers to verify the planning process.
This process of inclusion and exclusion allowed us to focus the review on studies that contribute to understanding the role of phenolic compounds in fabaceous seeds about diabetes mellitus, avoiding research that was not aligned with the specific objectives of the review. This is marked in the figure, and thanks to your comment, it was added to the text (lines 82-89).
Human Studies:
A major shortcoming of the paper is the lack of inclusion of human studies. There are numerous studies on the effects of legumes on diabetes management in patients. These must be incorporated for a comprehensive review. The conclusion suggesting future perspectives on this topic is misleading, as such studies are already being conducted.
Thank you for your comment. Specific studies of phenolic compounds in Fabaceae have been investigated. Although not many have been found, your comment has broadened our perspectives by considering the possibility that these compounds may be toxic for human consumption.
Other Comments:
Line 47: Additional mechanisms should be mentioned.
Thanks for the recommendation. We have added the mechanism of action to complement the analysis.
Line 53: Rewrite this sentence for specificity—e.g., phenolic acids, flavonoids, and isoflavonoids should be explicitly mentioned.
Thank you for the information; we have incorporated your suggestion to complement the content.
Line 108: The manuscript fails to mention mechanisms involving the latest antidiabetic drugs, such as GLP-1 analogs. This omission needs rectification.
Thank you for the recommendation. We have added information about it.
https://pmc.ncbi.nlm.nih.gov/articles/PMC8085572/
Line 208: The statement regarding the inhibition of enzymes by phenolic compounds needs proper referencing.
Thank you for your comment; the reference has been added in the indicated sentence.
Line 289: Clarify the impact of γ-aminobutyric acid (GABA) on diabetes management. Provide relevant studies and data to support its inclusion.
Thank you for your comment. That part has been rewritten to make the information presented clearer.

Reviewer 3 Report
Comments and Suggestions for Authors
The manuscript shows potential of legumes in handling Diabetes not only as a life style (dietary) modification but also in additional (supplementary) treatment. Technical issues: sections 3,4, and 5 should be placed before methods section as they are parts of introduction to the main subject: influences of fabeceous edible plants on diabetes. Conclusion should be section 8th. Merit issues:
line 59: should be prevent cell damage not attack cell damage;
line 61: should be "present commonly in fabeceous..."
Figure 1: 782 record were found (without doublets), but only 230 were screened- why? explain and supplement the main text;
102-107: the main cause of T2D is insulin resistance and secondary is impairment in insulin secretion as a effect of resistance and inflammation caused by hyperglycemia; T1D is primary caused by failure of beta -cells in insulin production and liberation due to they apoptosis; please correct this paragraph;
section 6: daidzin and genistein posses confirmed estrogenic properties - authors should mention this as estrogens influence diabetes (Alfonso-Magdalena P. Nat Rev Endocrinol 2011, 7: 346-353. Mouvais-Jarvis F. Endocr Rev 2013, 34: 309-338);
lines 286-289: give reference;
Author Response
The manuscript shows potential of legumes in handling Diabetes not only as a life style (dietary) modification but also in additional (supplementary) treatment. Technical issues: sections 3,4, and 5 should be placed before methods section as they are parts of introduction to the main subject: influences of fabeceous edible plants on diabetes. Conclusion should be section 8th. Merit issues:
Thank you for your comments. They were considered, and the changes were made.
Line 59: should be prevent cell damage not attack cell damage;
Thank you for your comment; the correction has been made.
line 61: should be "present commonly in fabeceous..."
Thank you for your comment; the correction has been made.
Figure 1: 782 record were found (without doublets), but only 230 were screened- why? explain and supplement the main text;
Thank you for your observation, what was marked in the figure as record screening was added to the text (lines 93-95).
102-107: the main cause of T2D is insulin resistance and secondary is impairment in insulin secretion as a effect of resistance and inflammation caused by hyperglycemia; T1D is primary caused by failure of beta -cells in insulin production and liberation due to they apoptosis; please correct this paragraph;
Thank you very much for your comment; we have incorporated the information you provided.
Section 6: daidzin and genistein possess confirmed estrogenic properties – authors should mention this as estrogens influence diabetes (Alfonso-Magdalena P. Nat Rev Endocrinol 2011, 7: 346-353. Mouvais-Jarvis F. Endocr Rev 2013, 34: 309-338);
We appreciate your comment. However, we have not specified what you mentioned, as it does not fit the article’s focus.
lines 286-289: give reference;
Thank you for your comment. The requested reference has been added.

Round 2
Reviewer 1 Report
Comments and Suggestions for Authors
The quality of the manuscript improved